# Isolation and Characterization of Spherical Cellulose Nanocrystals Extracted from the Higher Cellulose Yield of the Jenfokie Plant: Morphological, Structural, and Thermal Properties

**DOI:** 10.3390/polym16121629

**Published:** 2024-06-08

**Authors:** Solomon Estifo Wossine, Ganesh Thothadri, Habtamu Beri Tufa, Wakshum Mekonnen Tucho, Adil Murtaza, Abhilash Edacherian, Gulam Mohammed Sayeed Ahmed

**Affiliations:** 1Department of Mechanical Engineering, Adama Science and Technology University, Adama 1888, Ethiopia; solestifo074@gmail.com (S.E.W.); eba.almaz63@gmail.com (H.B.T.); 2Department of Materials Science and Engineering, Adama Science and Technology University, Adama 1888, Ethiopia; 3Faculty of Science and Technology, University of Stavanger, 4036 Stavanger, Norway; wakshum.m.tucho@uis.no; 4MOE Key Laboratory for Nonequilibrium Synthesis and Modulation of Condensed Matter, State Key Laboratory for Mechanical Behavior of Materials, School of Physics, Xi’an Jiaotong University, Xi’an 710049, China; adilmurtaza80@gmail.com; 5Mechanical Engineering Department, College of Engineering, King Khalid University, Abha 61421, Saudi Arabia; edalheriad@kku.edu.sa; 6Center of Excellence (COE) for Advanced Manufacturing Engineering, Department of Mechanical Engineering, Adama Science and Technology University, Adama 1888, Ethiopia; gmsayeed.ahmed@astu.edu.et

**Keywords:** cellulose, nanocellulose, thermal stability, crystallinity, morphological analysis

## Abstract

Scholars are looking for solutions to substitute hazardous substances in manufacturing nanocellulose from bio-sources to preserve the world’s growing environmental consciousness. During the past decade, there has been a notable increase in the use of cellulose nanocrystals (CNCs) in modern science and nanotechnology advancements because of their abundance, biocompatibility, biodegradability, renewability, and superior mechanical properties. Spherical cellulose nanocrystals (J–CNCs) were successfully synthesized from Jenfokie micro-cellulose (J–MC) via sulfuric acid hydrolysis in this study. The yield (up to 58.6%) and specific surface area (up to 99.64 m^2^/g) of J–CNCs were measured. A field emission gun–scanning electron microscope (FEG-SEM) was used to assess the morphology of the J–MC and J–CNC samples. The spherical shape nanoparticles with a mean nano-size of 34 nm for J–CNCs were characterized using a transmission electron microscope (TEM). X-ray diffraction (XRD) was used to determine the crystallinity index and crystallinity size of J–CNCs, up to 98.4% and 6.13 nm, respectively. The chemical composition was determined using a Fourier transform infrared (FT–IR) spectroscope. Thermal characterization of thermogravimetry analysis (TGA), derivative thermogravimetry (DTG), and differential thermal analysis (DTA) was conducted to identify the thermal stability and cellulose pyrolysis behavior of both J–MC and J–CNC samples. The thermal analysis of J–CNC indicated lower thermal stability than J–MC. It was noted that J–CNC showed higher levels of crystallinity and larger crystallite sizes than J–MC, indicating a successful digestion and an improvement of the main crystalline structure of cellulose. The X-ray diffraction spectra and TEM images were utilized to establish that the nanocrystals’ size was suitable. The novelty of this work is the synthesis of spherical nanocellulose with better properties, chosen with a rich source of cellulose from an affordable new plant (studied for the first time) by stepwise water-retted extraction, continuing from our previous study.

## 1. Introduction

Cellulose is the most common organic polymer on Earth and is not soluble in water, and 10^10^–10^11^ tons are produced annually [1,2]. The linear homopolysaccharide known as cellulose is composed of D-glucopyranose units joined by β-1, 4-linkages. Each glucose residue has three hydroxyl groups that are crucial for cellulose molecules to organize into stable crystals over time [3]. The crystalline polymorphs of native cellulose, known as cellulose I, II, III, and IV, each have a unique packing arrangement. Most likely, only type I (Iα and Iβ) cellulose exists in nature; types II, III, and IV are produced chemically from type I cellulose [4]. Most biomass materials, including wood and agricultural waste products, as well as marine algae, bacteria, fungi, and tunicates, are sources of cellulose [5]. Cellulose is a naturally occurring material that is biodegradable, renewable, and non-toxic [6].

Nanocellulose comprises cellulose particles with a minimum of one nanoscale dimension, ranging from 1 to 100 nm [7]. Being so prevalent in the plant world, cellulose polysaccharide units occur principally in three hierarchal (nano) structural forms, namely, cellulose nanocrystals (CNCs) (stiff and short), cellulose nanofibrils (CNFs) (flexible and long), and bacterial nanocellulose (BNC) (crystalline and pure) [8,9]. The extremely crystalline CNC particles have a width of 3–50 nm and a length of 100 nm to several μm. A CNF has a long structure with both amorphous and crystalline areas, ranging in width from 5 to 100 nm and length up to 100 μm [10,11,12]. BNC (4–50 nm width and >500 nm length) is also one of the main types of nanocellulose which is produced by a bacterial species known as Gluconoacetobacter xylinium using a bottom-up method from glucose molecules [13]. The origin of the cellulose, the method of extraction, the application, and its crystallinity or fibrousness all affect the qualities of nanocellulose [14].

The method used, the conditions of preparation, and the source of cellulose substance all affect the morphology, geometrical parameters (length, diameter, thickness, aspect ratio), yield, and crystallinity of CNCs. The morphology, geometrical dimensions (length, diameter, thickness, aspect ratio), yield, and crystallinity of CNCs depend on the method employed, the preparation condition, and the cellulose substance’s origin. Sulfuric, hydrochloric, phosphoric, and hydrobromic acids—or a combination of inorganic and organic acids—are the acids that are most frequently employed to produce CNCs. The most widely used method to create CNCs is generally the hydrolysis of cellulose with 60–64% sulfuric acid at a temperature range of 45 to 60 °C [1,15]. Anionic sulfate half-esters (–OSO_3_^–^) on the surface of the CNCs provide electrostatic stability and make the aqueous suspensions of CNCs produced by sulfuric acid catalyzed hydrolysis colloidally stable. On the other hand, because of the low-density surface charges, the ones that are removed via HCl hydrolysis show restricted colloidal stability [16].

Nanocellulose (NC) has unique properties of a large surface area, high strength, high elastic modulus, biodegradability, biocompatibility, sustainability, lightweight, high level of polymerization, ability to facilitate surface functionalization, chemical resistance, and high crystallinity. Numerous prospective applications have made use of these features [17,18]. Reinforcing nanocellulose in the polymer matrix can improve the strength, stiffness, toughness, thermal, barrier, antibacterial, and antioxidant properties of nanocomposites compared to neat polymers. Furthermore, by adding just a small amount of nanocellulose, the characteristics of the nanocomposite were greatly improved because of the fillers’ enormous surface area. The network of hydrogen bonds that cellulose forms inside the polymer matrix is what causes this phenomenon [19,20,21]. Nanocellulose has a multitude of potential uses, including impregnated textiles, packaging, biomedical, paper and pulp, coatings and paints, electronics, composites, water filters, etc. [19,22].

The Jenfokie climber plant has rapid shoot growth and a long internode, the least stem and leaf area ratio compared to erect plants, vast vessels to carry more water up the stem, and slow development and expansion of the leaf [23]. The plant has weak and flexible solid stems and separate internodes, releases a milky-like liquid substance (during harvesting), and grows rapidly by winding around a support (trees or green wall), creating large arcs [24,25]. The plant is non-flowering and has fibers between the internode along the entire stem from the bottom tip to the apex, which has not been used for different engineering purposes so far except for fencing in Ethiopia. The fibers extracted from the Jenfokie plant stem run along the entire stem and have a very long height (18 m) compared to other bast plants like sisal (1.2–1.5 m), false banana (4–13 m), stinging nettle (2–3 m), flax (0.9–1.2 m), and jute (2.73 m), and this would be a great way to obtain a large ratio of fiber to volume [26,27,28,29,30]. Water-retted, isolated Jenfokie cellulose (greater yield) was thus used in this investigation to synthesize and characterize nanocellulose, which was developed from our previous study [31].

The previous study focused on the extraction of fibers and cellulose from two plants with two extraction methods, while this study focused on nanocellulose synthesis by further treating and choosing the one with a high cellulose content and better properties. This study aims to synthesize and characterize nanocellulose from the highest yield of cellulose, crystallinity, and tensile strength of cellulosic fiber found in our previous analysis of a water-retted, extracted source and an unstudied (new) plant source. The study further investigated the properties of crystalline nano-sized cellulose, which had better properties than cellulose. Cellulose nanocrystals were synthesized in this study after the bleaching process using sulfuric acid hydrolysis and sonication. The yield, shape, particle size, specific surface area, morphology, crystallinity, functional groups, and thermal characteristics of cellulose nanocrystalline were also examined in this study. Further treating the cellulose obtained in our previous study with a combination of hydrogen peroxide and alkali increased the crystallinity of the microcellulose. The spherical nanocellulose found in this study with a high crystallinity and a large specific surface area could be useful for manufacturing nanocellulose polymer-reinforced bio-nanocomposites for the next study.

## 2. Materials and Methods

### 2.1. Raw Materials

The water-retted, extracted cellulose of Jenfokie found in our previous work [31] was used in this study for further cellulose nanocrystalline preparations. Chemicals such as ethanol absolute (99.8%), sodium hydroxide pellets LR (98%), sulfuric acid (assay: 98%), magnesium sulfate heptahydrate (MgSO_4_·7H_2_O), and hydrogen peroxide LR (H_2_O_2_, 3 and 30% solution) were bought from Fine Chemical General Trading Addis Ababa, in Ethiopia.

### 2.2. Micro- and Nanocrystalline Cellulose Preparation

The retted, extracted Jenfokie cellulose (higher yield and crystallinity) obtained in our previous study was further treated using 30% H_2_O_2_ (*v*/*v*), 2% NaOH (*w*/*v*), and 0.3% MgSO_4_·7H_2_O (*w*/*v*) (as a stabilizer) at 90 °C for 30 min, keeping the solid-to liquid ratio 1:10, as shown in Figure 1. Then, the sample was washed using distilled water until it reached a neutral pH and dried in an oven. In the second stage, the treatment was repeated, keeping the above solvent ratio the same by changing only the concentration of H_2_O_2_ to 3% and washing with distilled water. The most popular technique, acid hydrolysis, was then used, utilizing 64 wt.% H_2_SO_4_ and maintaining a solid-to-liquid ratio of 1:10 for 40 min at 45 °C while mechanically stirring to prepare the nanocellulose [32]. The reaction mixture was diluted ten times by volume with cold distilled water to stop the reaction. The excess acid was then removed from the suspension by triple centrifuging it for 15 min at 5000 rpm, adding ethanol. Once the sample reached a pH of neutral, it was dialyzed against distilled water for a week while the water was changed per day by manual stirring with the help of a glass rod. To prevent overheating, the CNC suspension was then sonicated for 15 min in an ice bath. After the process, the obtained CNCs were oven dried at 50 °C. Micro-cellulose was produced through mechanical grinding using an electric-driven dry grinding machine, which was sieved to obtain an average particle size in the range of 7.2–19.3 μm.

### 2.3. Characterization of Nanocellulose

#### 2.3.1. Yield Measurement of J–CNC

Once dialysis was completed, the total volume of J–CNC suspension was determined. A preset volume (mL) of the J–CNC suspension was transferred to a weighing bottle and dried at 105 °C in order to preserve a constant weight. After cooling to room temperature, the sample was weighed using an analytical balance [33]. The mean of three observations was used to obtain the final result for each sample. The yield was computed using Equation (1):(1)$$Yield(\%)=\frac{{(m}_{1}-m_{2})\times V_{1}}{m_{3}\times V_{2}}\times 100$$
where mass bottle (mg) and oven-dried J–CNC are represented by m_1_, the mass of the bottle (mg) is m_2_, J–MC weight (mg) is represented by m_3_, J–CNC suspension volume (mL) by V_1_, and oven drying volume (mL) of J–CNC by the number V_2_.

#### 2.3.2. J–CNC’s Specific Surface Area

The specific surface area [34] of the resulting spherical nanocellulose was computed with the subsequent Equation (2):(2)$$Specific\;surface\;area(SSA,\frac{m^{2}}{g})=\frac{6}{\rho \times d}$$
where d = average diameter of J–CNC, and ρ = density of cellulose.

#### 2.3.3. Morphological Properties

Both SEM and TEM experimental analyses were conducted at the University of Stavanger, Norway. Gemini SUPRA 35VP (ZEISS) (Carl Zeiss, Jena, Germany) with an EDAX energy-dispersive X-ray spectroscopy (EDS)-equipped field emission gun–scanning electron microscope (FEG-SEM) was used to characterize the morphology of J–CNC and J–CNC samples coated with gold (70 nm thick). The J–CNC sample’s morphology and dimensions were examined in more detail using a 200 kv Transmission Electron Microscope (TEM) JEOL2100 (JEOL, Tokyo, Japan). The sample was dissolved in ethanol and ultrasonicated for about 20 min and one drop was deposited on a holey carbon copper grid. Using digital image analysis (ImageJ software 1.53e), the sample’s average diameter was determined, confirming that it included both micro- and nano-sized cellulose. With the aid of OriginPro 2021 and ImageJ software, the particle size distribution was assessed on a TEM picture.

#### 2.3.4. Fourier Transform Infrared (FT–IR) Spectroscopy Analysis

The FT–IR analysis of water-retted, extracted Jenfokie cellulose (J–MC) and cellulose nanocrystals (J–CNCs) was examined using KBr pellets and the Spectrum 65 FT-IR (PerkinElmer, Waltham, MA, USA) in the 4000–400 cm^−1^ range.

#### 2.3.5. Thermogravimetric Analysis (TGA)

Using a thermogravimetric analyzer (Shimadzu DTG-60H, Shimadzu, Corp., Kyoto, Japan) with a nitrogen environment and a flow rate of 50 mL/min, the thermal stability of J–MC and J–CNC was assessed from ambient temperature to 700 °C, at a heating rate of 15 °C/min.

#### 2.3.6. Crystallinity Index by X-ray Diffraction (XRD)

J–MC and J–CNC samples were examined using an X-ray diffractometer (XRD-7000S, Shimadzu, Japan) equipped with a Cu Kα radiation source running at 30 mA and 40 kV. The diffractometer was continuously scanned in the 2θ range between 5 and 65° at a speed of 2°/min. Using the following Equation (3), the crystallinity index (CrI) of each sample was determined using Segal’s peak height method after baseline correction [35].
(3)$$Crystallinity\;index(\%)=\frac{I_{002}-I_{am}}{I_{002}}$$
where I_002_ represents the highest level of intensity crystal plate (002), at 2θ = 22.5° (J–MC), 2θ= 21.9° (J–CNC); I_am_ the highest level of intensity between 200 and 110 peaks (at 2θ = 20.3°, 2θ = 19.9° for J–MC, and J–CNC, respectively) which represents the amorphous mater.

The average crystallinity size (D_hkl_) was measured using the Scherer Equation [36] and Equation (4):(4)$$D_{hkl}=\frac{0.9\lambda }{\beta cos\theta }$$

In this case, D_hkl_ denotes the crystallite diameters of the miller’s diffracting planes, which serve as an indicator for hkl, 0.9 describes the correction factor of k, and the wavelength of radiation is denoted by λ (0.15406 nm), while the full width at half of the maximum height (FWHM) of the deconvoluted crystalline peak is represented by β in radians, and the diffraction angle expressed in radians is denoted by θ.

## 3. Results and Discussion

### 3.1. Analysis of J–CNC Yield and Specific Surface Area

For each step, 5 g of J–MC was taken and hydrolyzed with 64 wt.% sulfuric acid. The yield of J–CNC was measured between 54 and 61%, relating the initial weight of J–MC to the final weight of J–CNC. Three measurements were used to calculate the J–CNC 58.6% average yield. This result is in line with other studies such as nanocrystalline cellulose from *Pennisetum hydridum* (43.6%), surgical cotton (56%), pistachio shell (77.1%), and cotton (74.2%) [37,38,39,40]. The specific surface area of J–CNC in this study is found to be 99.64 m^2^/g. The result obtained in this study is greater than the high specific surface area of spherical nanocellulose obtained in other studies, such as cellulose nanocrystal from polar wood (24.2–28.9 m^2^/g) and agricultural waste (82.8 m^2^/g) and less than the spherical nanocellulose of tunicate (122 m^2^/g), rice husk (498.5 m^2^/g), and Agave teqiulana Weber (150 m^2^/g) [19,41,42,43].

### 3.2. Morphological Properties

#### 3.2.1. Scanning Electron Microscope

The hydrogen bonds within and between molecules in cellulose fibers have been reported to cause the commonly used cellulose, composed of areas that are crystalline and amorphous. To obtain CNCs, the amorphous areas can be eliminated by acid hydrolysis. The morphologies of J–MC and J–CNC from SEM micrographs are presented in Figure 2a,c. From the analysis, the diameter of J–MC was in the range of 7 to 19 μm (average diameter of 12.8 μm) and several micrometers in length. The particle size obtained in the J–MC sample in this study is in line with the previous study [44]. Figure 2b shows that the J–MC had comparatively homogeneous size distributions (*p* < 0.05).

#### 3.2.2. Transmission Electron Microscope

The J–CNC was hydrolyzed in sulfuric acid and then separated into nano-sized cellulose. The drying stage of sample preparation resulted in the appearance of aggregates, which were seen in the TEM micrographs along with spherical-shaped nanocrystalline cellulose, as shown in Figure 3a–f. This spherical shape is consistent with the prepared nanocellulose. From the TEM analysis, the mean particle size of J–CNC 34 nm was measured. This result is in line with the result noted [45,46]. Because of the van der Waals force, stacking nanocrystals may be the cause of the observed spherical shape morphology [47]. J–CNC exhibited the nanoscale size in their TEM images, which were prone to form agglomerates among CNCs attributed to hydrogen bonding because several hydroxyls are present, as shown in Figure 3e. Strong intermolecular hydrogen bonds between the particles and possible hydrophilic interactions between the chains may have contributed to the agglomeration, which was also observed by [36,48,49,50]. The TEM-produced J–CNC particle size distribution histograms, which range in size from 10 to 93 nm and are not statistically different from a population with a normal distribution (*p* > 0.05), are displayed in Figure 3f.

The crystalline structure of the J–CNC nanosphere taken along the zone axis using a transmission electron microscope with a high resolution (HR–TEM) is shown in Figure 4, as well as its associated fast Fourier transform (FFT) and inverse fast Fourier transform (IFFT) pattern. Furthermore, the crystalline lattices of J–CNC and the chosen region diffraction pattern were also investigated. It was determined that the inter-planar spacing (d-spacing) of the prevailing HR–TEM fringe was 0.33 nm between the vertical black and white lines. XRD d-spacing measurements (*d* = 0.45 nm) were taken to compare the lattice fringes and crystal structure of the J–CNC nanosphere with the HR–TEM data.

### 3.3. Fourier Transform Infrared (FT–IR) Spectroscope Analysis

FTIR spectrum of micro-cellulose and extracted nanocellulose are shown in Figure 5. There are two primary absorption regions observed at the highest wave number (3418–3051 cm^−1^) and at the lowest wave number (1741–604 cm^−1^). In the J–MC and J–CNC samples, the OH bending vibration of the water absorbed was linked to the peaks in the FTIR spectrum represented at 1636 and 1664 cm^−1^, respectively. The peaks in the 3418 cm^−1^ region of the J–MC sample are a result of α-cellulose’s O–H stretching, which has strong intra- and intermolecular H bonding [51]. J–MC also displayed the typical peaks at 2918 cm^−1^ due to the cellulose’s C–H stretching vibration. The acetyl group in the cellulose’s amorphous area caused the formation of a transmittance small peak at 1741 cm^−1^, which vanished after acid hydrolysis [52]. In support of cellulose’s presence, the peaks at 1432 cm^−1^ were caused by CH_2_ bending in symmetry, at 1375 cm^−1^ by –CH bending, and 1055 cm^−1^ by C–O stretching vibration in the J–MC sample [53,54]. The peak appearing at 607 cm^−1^ for J–MC was due to OH cellulose that has bent out of plane [10].

The strong dominant absorbance peak in the region of 3051 cm^−1^ is brought about by the hydroxyl (O–H) groups in the J–CNC sample bending and hydrogen bonds extending in the cellulose structure [55]. The peak represented in the 2507 cm^−1^ band is a result of the C–H stretching vibration of cellulose [56]. Cellulose is responsible for both the glucose ring stretch and the C–O vibration of crystalline cellulose, with the J–CNC peak being recorded at 1108 cm^−1^ [35]. Conversely, a sharp peak like this enhanced the nanocellulose’s surface area and was indicative of pure cellulose [57,58]. The peak at 914 cm^−1^ in the J–CNC sample was caused by the cellulose’s C–H rocking vibration and glycosidic linkages to the glucose ring [48,59]. The β-glycoside linkage glucose ring of cellulose is responsible for the bands at 894 cm^−1^ that were observed for J–MC [60,61]. The peaks overserved at 604 cm^−1^ for the J–CNC sample were due to the OH out-of-plane bending of cellulose [53].

### 3.4. Thermogravimetric Analysis (TGA)

Figure 6 shows the thermal gravity analysis (TGA), derivative thermograms (DTGs), and thermogravimetry analysis (TGA) curves and temperatures at which the loss of mass rate of the J–MC and J–CNC samples occurs. Table 1 contains data on the mass loss, onset temperature, and temperatures with the maximum rate of degradation. Two major stages of weight loss were experienced by J–MC ranging from 234 to 375 °C and 375 to 536 °C, while the weight loss of J–CNC was found in the range of 100–258 °C and 258–529 °C. The first mass loss of J–MC (7%) from ambient temperature to 101 °C and of J–CNC (9%) from room temperature to 100 °C in the first region was ascribed to water evaporating and low molecule compounds [62]. At 700 °C, the J–MC and J–CNC experienced thermal weight loss rates of 97% and 65%, respectively.

The ending temperature of region I (onset temperature at 234 °C) for the second degradation temperature range of approximately 234 to 375 °C (*T*_max_ = 366 °C) of the J–MC sample, and the onset temperature at 100 °C for the second degradation temperature in the range 100–258 °C (*T*_max_ = 133 °C) of the J–CNC sample, was observed from the TGA/DTG curves. A significant weight loss of 54 and 37% for micro-cellulose and nanocrystalline cellulose is observed in this region, respectively. Glycosyl structure breakdown and depolymerization are involved in the second degradation in this area, which is linked to cellulose pyrolysis [63,64]. Spherical J–CNCs at low temperature in this region resulted from the amorphous degradation area, which were more accessible and sulfated [65]. The initial thermal decomposition of J–CNCs (100 °C) occurred earlier during the second pyrolysis stage than that of J–MC (234 °C), demonstrating the increased thermal stability of the J–MC [66]. However, the thermal degradation in the range 186–400 °C of the plateau region of J–CNC indicates good thermal stability. These findings could be due to the carboxyl groups of J–CNCs and the residual anionic sulfate half-ester groups, which resulted in a decline in thermal stability [67]. Additionally, nanocellulose’s small size (less than 100 nm) and spherical shape resulted in an extraordinarily vast surface that comes into direct contact with high temperatures [58].

In region III (onset temperature at 395 °C), the third degradation temperature between about 395 and 536 °C (*T*_max_ = 475 °C) of the J–MC sample is attributed to cellulose’s depolymerization to produce volatiles, while the onset temperature at 405 °C and the third degradation temperature in the range of 258–529 °C (*T*_max_ = 445 °C) for that of the J–CNC sample are attributed to the un-sulfated crystalline region and subsequently charring degradation of carbonic residues into components of the gaseous product with a lower molecular weight [68,69,70]. Comparing isolated J–CNCs to J–MC, the former has less thermal resistance. This could be caused by the presence of acid groups and a larger surface of nano dimensions that are exposed to more heat [71]. The thermal stability increases as the surface charge density decreases. The increase in density of sulfate half-esters is linked to the reduced thermal stability of CNCs in multiple studies [72]. The residue mass of J–MC (3%) and J–CNC (35%) was observed at 700 °C. J–CNC exhibited a greater char residue than the J–MC sample because of the inclusion of flame-retardant sulfate ester groups [73]. The high char yield obtained in this study is in line with nanocellulose obtained from pine cellulose using sulfuric acid hydrolysis [65].

DTA, along with TGA and DTG, also certified the thermal stability of J–MC and J–CNCs. The DTA study also determined the nature of the created crystals and their thermal resistance, highlighting the crystal’s potential in an endothermic or exothermic fashion. The endothermic process occurred at 23–141 °C for J–MC and 23–263 °C for J–CNC, while the ectothermic process occurred at 141–556 °C for J–MC and 263–497 °C for J–CNC. The first endothermic peak values occurred at temperatures below 100 °C, indicating moisture loss and evaporation in both the J–MC and J–CNC samples in the DTA curves. The second endothermic process, in which the J–CNCs broke down at 130 °C, is characterized by the crystallite melt that reveals the breakdown processes and provides evidence for TG/DTG testing. Acid hydrolysis caused the nanocrystals’ surface to become less thermally stable, which led to the maximum breakdown occurring at low temperatures as a result of sulfate ion attachment [74]. The more thermally stable and exothermic third peak occurred at temperatures of 334 and 462 °C for the J–MC and at a temperature of 444 °C for the J–CNC samples, respectively, indicating the depolymerization of glycosyl units and the formation of carbonaceous residues [75]. There are several reasons why J–CNC depolymerizes, including (i) its nano-size and higher number of free ends, which break down at a reduced temperature; (ii) the presence of sulfuric acid, which makes cellulose easier to depolymerize by eliminating some hydroxyl groups; (iii) the abundance of H+ ions in the weakly acidic atmosphere, which increases the amount of char residue through the elimination of oxygen in the form of water, preventing weight losses; and (iv) the highly crystalline nature of J–CNC increases the number of carbon residues [76,77,78].

### 3.5. Crystalline Structure Analysis

XRD patterns of J–MC and J–CNC samples are represented in Figure 7. The J–CNC sample’s cellulose II diffraction peaks were attributed to cellulose II and are located at round 2θ = 12.2°, 2θ = 20.3°, and 2θ = 22.5°, respectively. These correspond to the (1$\bar{1}$0), (1 1 0), and (0 2 0) planes. The J–CNC shows three major characteristic peaks at around 2θ = 12.5°, 2θ = 19.9°, and 2θ = 21.9°, which correspond to the (1$\bar{1}$0), (1 1 0), and (0 2 0) planes, which are the crystalline phase of cellulose II [79]. Following bleaching and acid hydrolysis, the crystallinity index rises, most likely as a result of the elimination of amorphous substances like lignin, hemicellulose, and disordered amorphous areas (i.e., cellulose), respectively. These peaks are also present in the J–CNC, consistent with the findings of earlier research [80].

Table 2 lists the result of measurements made using X-ray diffraction curves to determine the mean diameter, mean d-spacing, and mean crystallite size from the three strongest peaks, as noted above. Segal’s method measured the crystallinity indexes of J–MC at 92.1% and J–CNC at 98.4% after baseline correction. The elimination of the amorphous components during hydrolysis with sulfuric acid and the reorganization of the crystalline regions into a more ordered structure are the causes of the increase in the crystallinity index for the J–CNC sample [81,82].

When the amorphous parts of microcellulose were removed during acid hydrolysis, the crystalline regions were reorganized into a better-ordered structure, which increased the J–CNC crystallinity index compared with J–MC [21,83,84]. The maximum crystallinity index of J–CNC found in this investigation (98.8%) is in line with other higher cellulose nanocrystals of Kombu that have been researched, including cotton filter (93.4%), rice straw (91.2%), South African cotton (SANC) (97.8%, Chinese cotton (92.4%), cotton cloth waste (95%), and sugarcane peel (SPCNC) (92.2%) [70,85,86,87,88]. Additionally, the creation of highly crystalline, stable nanocellulose is favored by a higher achieved crystallinity index [75]. Because of its high CrI, spherical nanocellulose may enhance the mechanical, thermal, and other properties of nanocellulose composites [44]. Another important factor associated with the cellulose crystal structure is the size of the crystallite. The process of recrystallization that exists in nanocellulose crystal structures increases in crystallite size [1].

The crystallinity index of J–CNC increased after sulfuric acid hydrolysis compared to the J–MC sample in this study, while the maximum thermal degradation of J–CNC, especially in region II, decreased compared to the maximum thermal degradation of J–MC in this region. Due to the existence of residual sulfate groups, which cause the dehydration process and ultimately lead to breakdown, the thermal stability of the nanocellulose produced by sulfuric acid hydrolysis is often lower than that of nanocellulose produced by other methods. [89,90,91]. It was thought that (i) the J–CNCs’ greater specific surface area, which is the result of their short nanocellulose chains, promotes the production of many free-end chains on the surface, which are more likely to break down quickly at lower temperatures since there has been greater subjection to a source of heat; (ii) nanocellulose degraded early after heating treatment because of its quick molecular weight loss during hydrolysis compared to its starting material; (iii) concentrated H_2_SO_4_ could harm crystalline domains in addition to removing loosely packed faulty sections, therefore increasing the molecules’ susceptibility to the degradation stage in response to temperature increases; and (iv) substitution of active sulfate groups (O–SO_3_H) for hydroxyl groups through esterification would start the dehydration reaction on the sulfated nanocellulose. As a result, water would be developed, which would accelerate the disintegration of the remaining cellulose into smaller pieces, while un-sulfated crystals tend to decompose at higher temperatures [91,92,93,94,95,96].

## 4. Conclusions

Cellulose nanocrystals were successfully synthesized from stepwise extracted micro- cellulose via a sulfuric acid hydrolysis and sonication process. Pretreatment, processing methods, and the supply of biomass from lignocellulosic plants will all affect the yield, surface morphology, shape, size, and other features of nanocellulose. Nanocrystals with a diameter of about 34 nm were discovered by TEM investigation of J–CNC at the nano-sized scale (below 35 nm). Such a small size, spherical-shaped nanoparticle will facilitate the interfacial interaction with polymeric materials during the reinforcing process, according to the results of the morphological investigation. A Fourier-transform infrared spectroscope also confirmed that the amorphous portion of cellulose was removed. From the XRD analysis, the crystallinity index and crystallinity size of the nanocellulose increased, which shows the successful removal of the amorphous region of the cellulose structure. Further pretreatment of the previously studied cellulose increased the crystallinity index due to the decrease (removal) of some of the non-cellulosic constituents that gave rise to pure microcellulose in this study. Spherical nanocrystals with a high crystallinity index were developed in this study based on the combined extraction effect, starting with the fiber preparation method, which would have wide potential and proven uses. When used to create nanocomposite materials, the high crystallinity index of nanocellulose may give them a high degree of stiffness. The thermal stability of nanocellulose is less than that of microcellulose resulting from the sulfuric acid treatment. The thermal stability of microcellulose also slightly decreased compared to cellulose extracted in our previous study, probably due to the removal of some of the remaining lignin (i.e., being more thermally stable). Although the maximum decomposition temperature of nanocellulose decreased, at 700 °C, the amount of residue left was greater than that of micro-cellulose. Since both the plant source and extraction method showed a profound effect on the properties of cellulose, as noted in our previous study, it was chosen to have the better cellulose and other properties (retted, extracted Jenfokie plant) for nanocellulose synthesis. The results obtained in this study also confirm that the processing conditions resulted in nanocrystals with better properties. The experimental output of this study therefore enables the introduction of a new (unstudied) plant that is sustainable, optimizes resources, is cost-effective, and has rapid growth in a wide range of environmental contexts as an alternative source of nanocellulose synthesized for a multitude of potential reinforcements in nanocomposites for applications spanning engineering, biomedical, packagings, paints, coatings, etc.

## Figures and Tables

**Figure 1 polymers-16-01629-f001:**
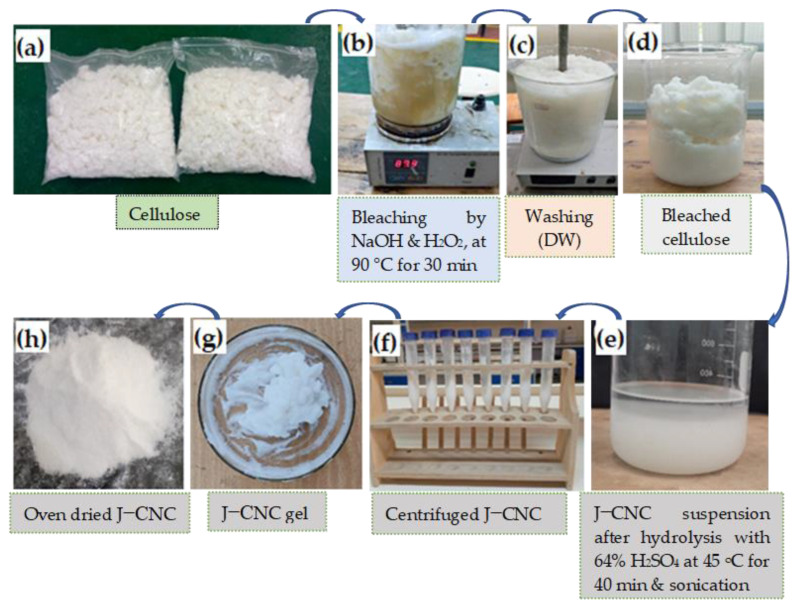
Flow chart of nanocellulose synthesis: (**a**) cellulose, (**b**) bleaching process, (**c**) washing, (**d**) bleached cellulose, (**e**) nanocellulose suspension, (**f**) centrifuged J–CNC, (**g**) J–CNC gel, and (**h**) oven dried J–CNC. DW: distilled water.

**Figure 2 polymers-16-01629-f002:**
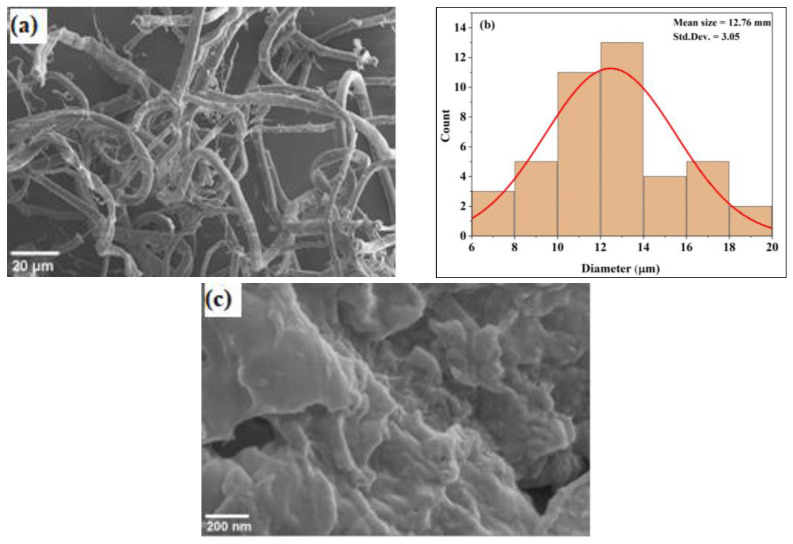
Scanning electron microscope micrographs of J–MC (478× magnification) (**a**) and diameter distribution histogram with normal distribution curve (red line) of J–MC (**b**) and J–CNC (50,000× magnification) (**c**).

**Figure 3 polymers-16-01629-f003:**
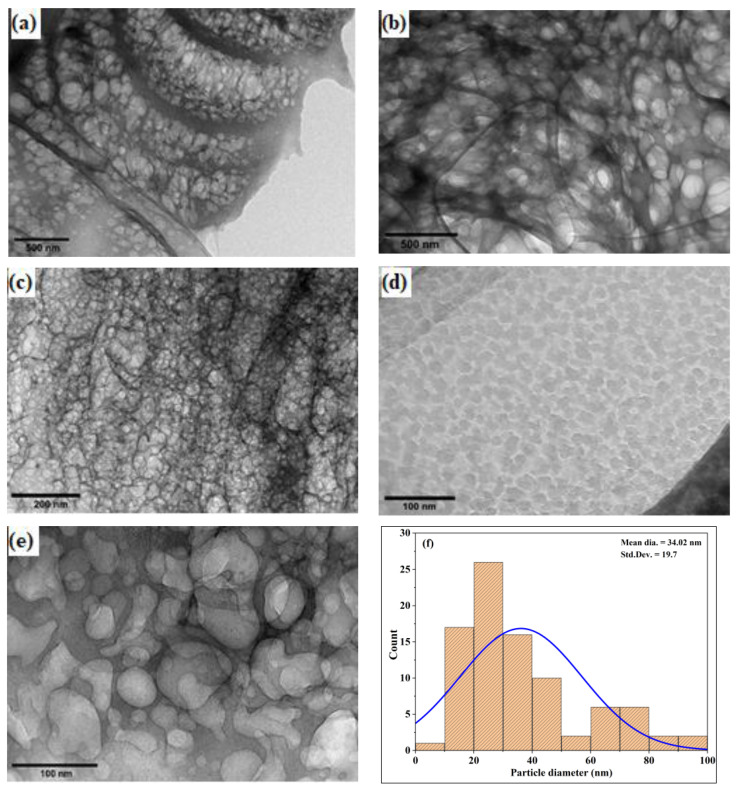
Transmission electron microscope morphology result of J–CNC (20,000× magnification) (**a**), (20,000× magnification) (**b**), (50,000× magnification) (**c**), (80,000× magnification) (**d**), (120,000× magnification) (**e**), and particle size distribution histogram with normal distribution curve (blue line) (**f**).

**Figure 4 polymers-16-01629-f004:**
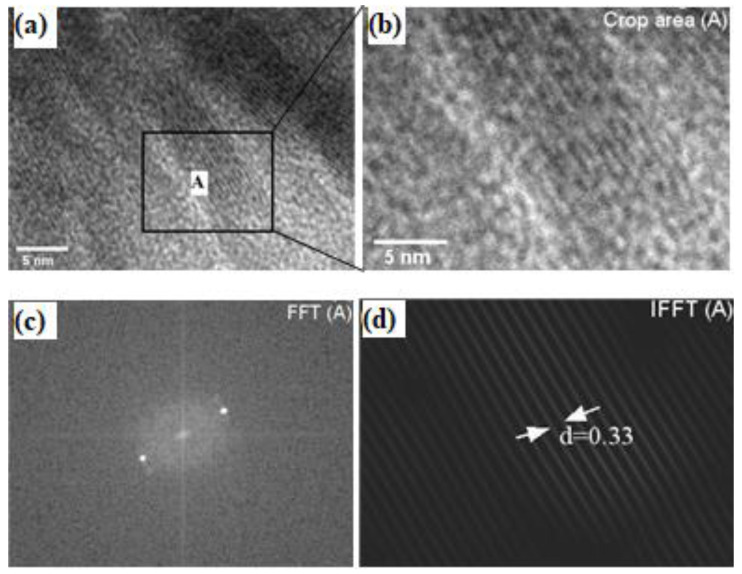
HR–TEM image of a single J–CNC nanosphere (**a**), specific area chosen for d-spacing analysis (**A**), selected area (**b**), and its corresponding FFT (**c**) and IFFT (**d**).

**Figure 5 polymers-16-01629-f005:**
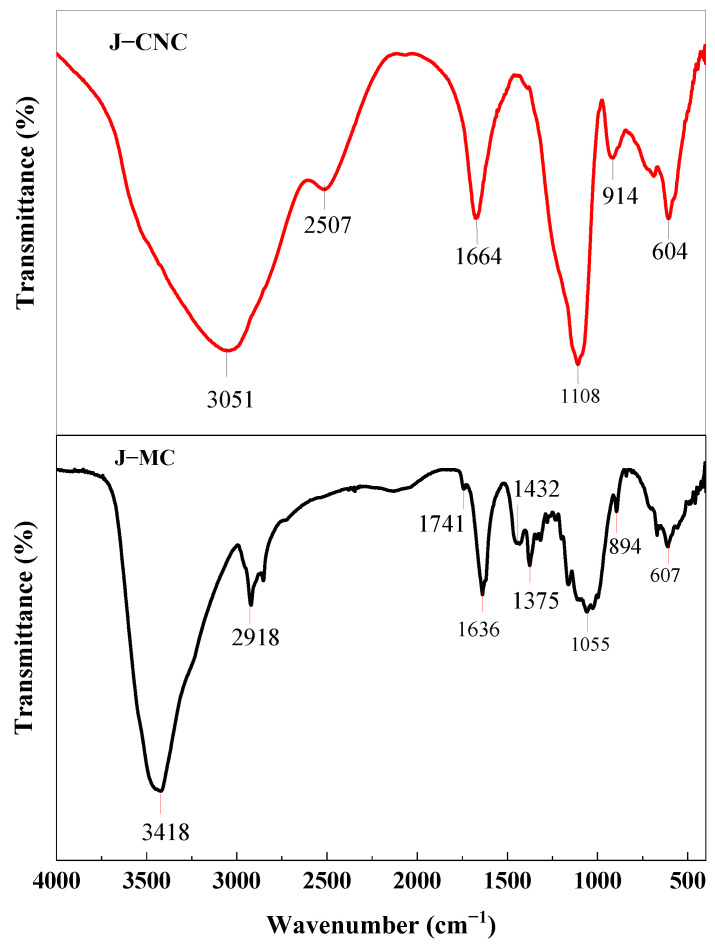
FT–IR analysis of J–CNC and J–MC samples.

**Figure 6 polymers-16-01629-f006:**
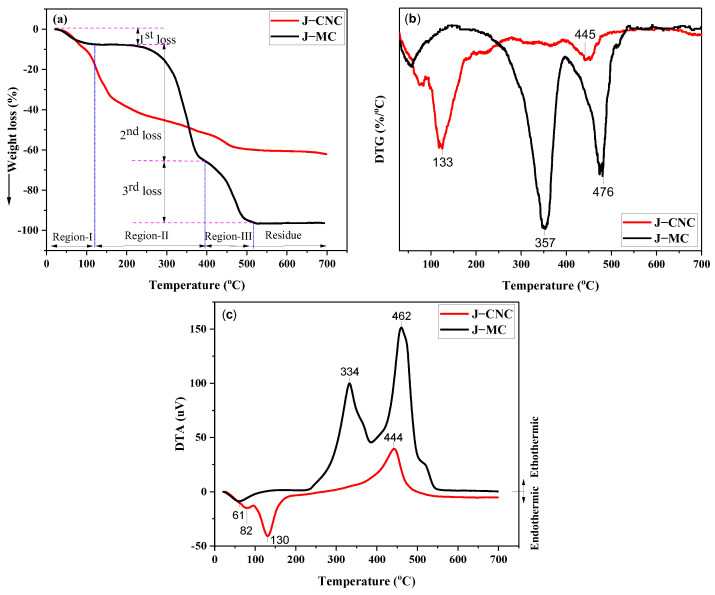
TGA, DTG, and DTA curves of the J–MC and J–CNC samples, respectively (**a**–**c**).

**Figure 7 polymers-16-01629-f007:**
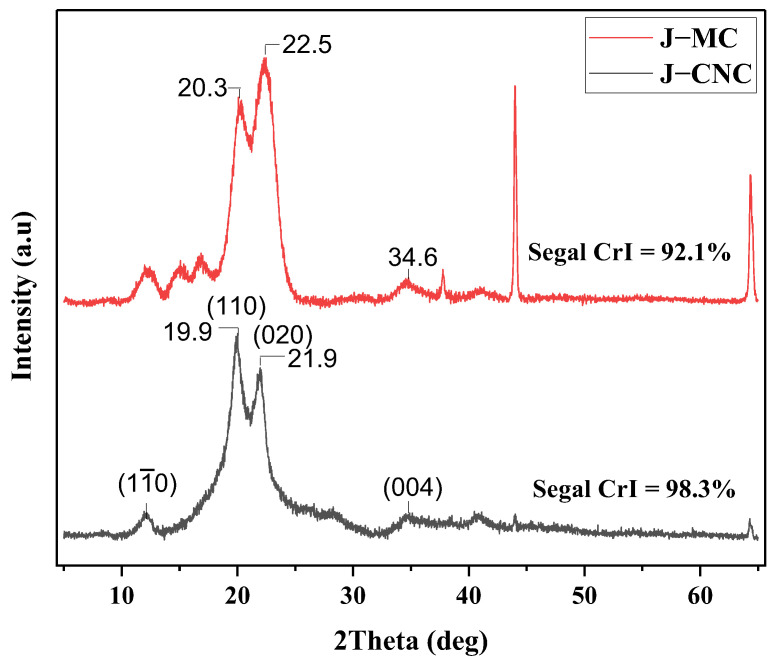
X-ray diffraction curves of J–MC and J–CNC samples.

**Table 1 polymers-16-01629-t001:** The onset temperature, the temperature on the highest rate of weight loss, and residual weight of J–MC and J–CNC.

Source	Region I	Region II			Region III			
	Temperature (°C)	*T*_onset_ (°C)	*T*_max_ (°C)	*WL* (%)	*T*_onset_ (°C)	*T*_max_ (°C)	*WL * (%)	Residue at 700 (°C)
J–MC	RT–123	234	357	54	395	475	35	3
J–CNC	RT–88	100	133	37	258	445	18	35

RT = room temperature, *T*_onset_ = degradation onset temperature, *T*_max_ = degradation temperature on maximum weight loss rate, and *WL* = weight loss.

**Table 2 polymers-16-01629-t002:** XRD results of d-spacing, crystallinity index, crystallinity size, and diameter of J–MC and J–CNC samples.

Samples	2theta (Degree)	d-Spacing (nm)	Crystallinity Index (%)	Mean Crystallinity Size (D, nm)	Mean Diameter J–MC (μm)	Mean Diameter J–CNC (nm)
J–MC	22.3	0.52	92.1	4.22	12.80	-
J–CNC	19.9	0.45	98.4	6.13	-	34

## Data Availability

The data will be made available on request from first author’s experimental PhD work conducted at Wachamo University and Adama Science and Technology University (ASTU) in Ethiopia.

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
