# Peer review of "Isolation and Characterization of Spherical Cellulose Nanocrystals Extracted from the Higher Cellulose Yield of the Jenfokie Plant: Morphological, Structural, and Thermal Properties"

_polymers, 2024, doi:10.3390/polym16121629_

Round 1
Reviewer 1 Report
Comments and Suggestions for Authors
The manuscript by Fossine et al. concerns the isolation of nanocellulose from a very specific plant, and its initial characterization.
The isolation process has no novelties (i.e. it is a kind of further step of something already published recently by some of the author) and the characterization was carried out by using methods commonly used for such purposes, such as TEM, SEM, XRD, TGA. The only point is that there are no reasons for including this content in a journal like polymers.
My suggestion is to improve it, following my minor comments, and then submit to a more appropriate journal.
- in the introduction, please cite some basic references instead of recent papers, especially when the topic is very general and well known since very long time
- at the end of the introduction, I would dedicate few more sentences to the content of the manuscript, to introduce what you did and why (and what are the differences in comparison whit the paper you recently published in another journal
- the beginning of section 4.1 is not clear enough, please what you are going to do (even if detailed in the experimental part) and then present and discuss the results
- in 4.2.2 do not report the decimal digits, it has no meaning (as the precision is much lower)
- in 4.4 do not report the decimal digits of the temperatures, as the equipments have not such precision
- conclusion: also highlight the differences between this new method and the old one. Moreover, stress the differences between this method applied to Jenfokie plant and the methods applied to other sources, or just say that all the method is nothing but the application of something already well known to another source …
Comments on the Quality of English LanguageIt has to be improved as many sentences loose part of their meaning
Author Response
Reviewer 1 – Comments and Response
We appreciate the time and effort put into providing feedback on our manuscript, as well as your comments and suggestions for improvements. Most of the suggestions were taken into account. The red underlining in the text word track highlights these modifications. These are detailed point-to-point responses to the comments and issues raised by reviewers. All page numbers (shown in blue) here refer to the revised manuscript file with revision history.
Reviewers’ Comments to the Author
Reviewer: 1
Q#1. In the introduction part, please cite some basic references instead of recent papers, especially when the topic is very general and well known since very long time.
Author response: Thank you!
It has been changed for some of the references for well-known and general topics and are incorporated in the text: (previous line 49, current line 51; previous line 52, current line 54; and previous line 54, current lines 56).
Q#2. At the end of the introduction, I would dedicate few more sentences to the content of the manuscript, to introduce what you did (and what are the differences in comparison with the paper you recently published in another journal.
Author response: Thank you!
The author incorporated what was done and some comparisons with the previous paper especially at cellulose level (previous lines 107-110, current lines 117- 124, and 127-131).
Q#3. The beginning of section 4.1 is not clear enough, please what you are going to do (even if detailed in the experimental) and then present and discuss the results.
Author response: Thank you for pointing that out!
The author incorporated the revised sentences (previous lines 201-202, current lines 222-225).
Q#4. In 4.2.2 do not report the decimal digits, it has no meaning (as the precision is much lower).
Author response: Thank you!
The decimal digits are corrected (previous line 32, current line 32; previous line 231, current line 255; previous line 239, current line 263; and previous line 406, current line 449).
Q#5. In 4.4 do not report the decimal digits of the temperature, as the equipment have no such precision.
Author response: Thank you for pointing that out!
The author corrected decimal digits of thermal properties (previous lines 296-299current lines 320-323; previous Figure 6 lines 301-304, current Figure 6 lines 325-327; previous line 301, current line 325; previous lines 305-309, current lines 329-333; previous lines 314-316, current lines 338-340; previous lines 322-325, current lines 346-349; previous line 333, current line 357; previous lines 340-341, current lines 364-365; previous lines 344, current line 368; previous lines 349, current line 373; and previous Table 1 line 359-360, current Table 1 line 383-384).
Q#6. Conclusion: also highlight the differences between the new method and the old one. More over stress, the differences between this method applied to Jenfokie plant and the methods applied to other sources, or just say that all the method is nothing but the application of something already well known to another source.
Author response: Thank you for pointing that out!
The details have been incorporated in the conclusion part (previous line 412, current lines 457-461; and previous line 425, current line 464-466 and 468-476).
Please check the attached word file for the same "Comments and Response"
Thank you for all of your fruitful comments and suggestions again.
Thanks

Reviewer 2 Report
Comments and Suggestions for Authors
The manuscript entitled ‘’ Isolation and Characterization of Spherical Cellulose Nanocrystals Extracted from the Higher Cellulose Yield of the Jenfokie Plant: Morphological, Structural, and Thermal Properties” is devoted to the production of nanocellulose from the Jenfokie plant. The authors used TEM, SEM, TGA, IR- spectroscopy and X-ray methods. The work cannot be called breakthrough. The authors have already published a paper on the extraction of cellulose from Jenfokie and its properties in 2023 [30]. The current study is a continuation of the work. The work contains all the required sections, the presentation style is clear. The conclusion is consistent with the results of the study.
Some questions arise after reading:
1. Please formulate the novelty more clearly (Abstract or Introduction).
2. Introduction: line 68-70 …depend depending…Please rewrite the sentence.
3. Line 91-101: The text has a very long description of the Jenfokie plant. Please shorten the paragraph.
4. Materials and Methods, TGA experiment: what is the heating rate of the samples? The authors did not indicate this in the method description.
5. Section Results and Discussion. The results are there, many experiments have been carried out, but the discussion is presented very briefly. The authors determined the crystallinity of MC and CNC. How does the crystallinity of samples affect thermal properties, for example, the temperature of maximum degradation? What is the observed relationship between the structure and properties of the resulting samples? The section should be summarized.
6. Figure 5. A letter is missing from the figure caption (J-C, please cheсk).
7. Line 385-387 …which increased the J–CNC crystallinity index compared with J–CNC? Please check.
8. The list of references is not compiled at the request of Polymers. The year of publication of the article should be indicated without brackets.
Comments on the Quality of English LanguagePlease use synonyms and correct mistakes.
Author Response
We appreciate the time and effort put into providing feedback on our manuscript, as well as your comments and suggestions for improvements. Most of the suggestions were taken into account. These are detailed point-to-point responses to the comments and issues raised by reviewers. All page numbers (shown in blue) here refer to the revised manuscript file with revision history.
Reviewers’ Comments to the Author
Reviewer: 2
Q#1. Please formulate the novelty more clearly (abstract or introduction).
Author response: Thank you for pointing that out!
The novelty of our study has been incorporated in the introduction part (previous line 42, current lines 42-44).
Q#2. Introduction: line 68-70 ...depend … depending please rewrite the sentence.
Author response: Thank you!
The sentence is corrected accordingly (previous lines 68-70, current lines 70-74).
Q#3. Line 91-101: The text has very long description of the Jenfokie plant. Please shorten the paragraph.
Author response: Thank you!
The author corrected accordingly in the text (previous lines 91-101, current lines 400-108).
Q#4. Materials and Methods, TGA experiment: what is the heating rate of the samples? The author did not indicate this in the method description.
Author response: Thank you for pointing that out!
The author incorporated heat-up rate (previous line 181, current line 202).
Q#5. Section results and discussion. The results are there, many experiments have been carried out but, the discussion is presented very briefly. The authors determined the crystallinity of MC and CNC. How does the crystallinity of samples affect the thermal properties for example, for example, the temperature of maximum degradation? What is the observed relationship between the structure and properties of the resulting samples? The section should be summarized.
Author response: Thank you for pointing that out!
The author incorporated how extraction affects crystallinity and thermal properties (previous lines 397, current lines 422-440).
Q#6. Figure 5. A letter is missing from the figure caption (J-C, please check).
Author response: Thank you for pointing that out!
The missed spelling is incorporated accordingly (previous line 290, current line 314).
Q#7. Line 385-387 … which increased the J-CNC crystallinity index compared with J-CNC? Please check.
Author response: Thank you!
The author corrected (previous line 385-187, current lines 409-411).
Q#8. The list of references is not compiled at the requisite of Polymer. The year of publication of the article should be indicated without brackets.
Author response: Thank you!
The author corrected the references accordingly.
Thank you for all of your fruitful comments and suggestions again.
Thanks

Round 2
Reviewer 1 Report
Comments and Suggestions for Authors
The authors took into account the suggested changes. On the other hand, as already stated, this is nothing but a well-known isolation process and has no novelties (i.e. it is a kind of further step of something already published recently by some of the author) and the characterization was carried out by using methods commonly used for such purposes, such as TEM, SEM, XRD, TGA. Moreover, and most important, I still believe that there are no reasons for including this manuscript in a journal like Polymers.
Comments on the Quality of English Languageminor corrections necessary
Reviewer 2 Report
Comments and Suggestions for Authors
The manuscript is revised. Accept in present form.